# Expressure: Detect Expressions Related to Emotional and Cognitive Activities Using Forehead Textile Pressure Mechanomyography

**DOI:** 10.3390/s20030730

**Published:** 2020-01-28

**Authors:** Bo Zhou, Tandra Ghose, Paul Lukowicz

**Affiliations:** 1German Research Center for Artificial Intelligence (DFKI), 67663 Kaiserslautern, Germany; paul.lukowicz@dfki.de; 2Department of Psychology, University of Kaiserslautern, 67663 Kaiserslautern, Germany; ghose@sowi.uni-kl.de; 3Department of Computer Science, University of Kaiserslautern, 67663 Kaiserslautern, Germany

**Keywords:** affective computing, smart textiles, pressure mechanomyography, facial expression, cognitive load, emotion recognition

## Abstract

We investigate how pressure-sensitive smart textiles, in the form of a headband, can detect changes in facial expressions that are indicative of emotions and cognitive activities. Specifically, we present the Expressure system that performs surface pressure mechanomyography on the forehead using an array of textile pressure sensors that is not dependent on specific placement or attachment to the skin. Our approach is evaluated in systematic psychological experiments. First, through a mimicking expression experiment with 20 participants, we demonstrate the system’s ability to detect well-defined facial expressions. We achieved accuracies of 0.824 to classify among three eyebrow movements (0.333 chance-level) and 0.381 among seven full-face expressions (0.143 chance-level). A second experiment was conducted with 20 participants to induce cognitive loads with N-back tasks. Statistical analysis has shown significant correlations between the Expressure features on a fine time granularity and the cognitive activity. The results have also shown significant correlations between the Expressure features and the N-back score. From the 10 most facially expressive participants, our approach can predict whether the N-back score is above or below the average with 0.767 accuracy.

## 1. Introduction

Over the last decade, physiological wearable monitoring has made tremendous advances, and more and more robust systems suitable for long-term everyday use are becoming available. Psychological monitoring, on the other hand, has been slower to make real-world impacts, mostly due to the fact that it requires more complex, often more subtle, sensing that is more difficult to implement in an unobtrusive, robust wearable form factor. Thus, for example, Galvanic Skin Response (GSR) is known to be a good indication of stress and cognitive load [1,2,3]. However, it relies on subtle electrical biosignals and requires careful placement of the electrodes and stable electrical contact to the skin. Similarly, while a rough estimate of the heart rate is sufficient for many physiological monitoring applications, psychological monitoring often relies on heart rate variability, which requires much more exact measurement [4,5,6].

In this paper, we propose and evaluate a novel wearable sensing system for the monitoring of emotional and cognitive states that are based on facial expression analysis. Detecting and analyzing facial expressions is a well-established research direction [7,8]. To date, most work in this area relies on computer vision techniques [9,10,11,12,13,14,15]. Methods with infrared cameras have also shown reliable results in recognizing emotional expressions [16,17,18]. While video-based face expression analysis methods have produced excellent results, they are unsuitable for wearable applications as they require a camera with a good view of the user’s face. Such a view is not easily achieved with a body-worn camera, and generally requires a stationary set-up in which the user sits or stands in front of a camera.

As an alternative suitable for wearable use, electromyography (EMG) has been studied [19,20] for monitoring facial muscle activity. Wearable electroencephalographic (EEG) has also emerged in recent years to detect brain activities [21,22,23]. While showing encouraging results, EMG and EEG suffer from the usual problems of all systems based on electrical biosignals: prone to noise and electrical-magnetic interference (EMI), associated placement requirements, and secure skin-electrode contact.

In this paper, we propose to use textile pressure mapping (TPM) arrays unobtrusively integrated into a headband to sense the mechanical deformation and muscle motions directly. It requires no electrical contact with the skin and, because of the array nature of the sensor, has no strict requirements on sensor placement (other than being placed roughly on the forehead). The approach, known as surface pressure mechanomyography (MMG), can be used both as an alternative sensing method to EMG or together with EMG to reduce errors and motion artifacts. While it has been well studied (see related work) on legs, arms and upper body muscles; there is very little work on using MMG for facial expression recognition. Our approach is based on two ideas:

(1) While emotions and cognitive states can form rich expression across various facial regions, much information is contained in the forehead and eyebrows area which can be easily covered with a headband (our approach) or glasses frames ([24,25]). Further, our headband approach does not conflict with the wearers’ own glasses.

(2) By using the textile pressure matrix, we can unobtrusively integrate an array of sensors into the headband that covers the entire relevant area. With appropriate processing methods, the system can be made relatively insensitive to placement variations and accommodate a broad range of different expressions related to forehead muscles. This is a significant advantage compared to systems based on individual force sensitive resistors (FSR) sensors (such as [24] and our abandoned early prototype).

### 1.1. Paper Contribution

In this paper, we present Expressure (EXpression detected by PRESSURE), a novel approach to using wearable textile MMG in the form of an unobtrusive wearable device (headband) to monitor cognitive and emotional states on the basis of facial expressions. We describe our system hardware and present a model to classify emotional states from the sensor data accurately. In two separate experiments motivated by standard psychology studies, we monitor the forehead muscle motions of participants (1) while they are mimicking emotional expressions and (2) while they are performing a high cognitive load task that evokes emotions.

In the first experiment, the participants mimicked visual stimuli displaying different eyebrow or facial expressions. The results validate that the Expressure approach can recognize between neutral, raised eyebrows, and lowered eyebrows with 0.824 accuracy (0.333 chance level), and 7 emotional expressions with 0.375 accuracy (0.143 chance).

In the second experiment, the participants performed N-back tasks, which introduce continuous working memory loads. We observed from video recordings that various forehead actions and head motions are usually related to emotions evoked by cognitively challenging stimuli. Based on statistical tests, we conclude that the data acquired and modeled by the Expressure approach have significant correlations (p<0.05) with the participant’s cognitive activity.

### 1.2. Paper Structure

Section 2 introduces the physiology and psychology background of this study. Section 3 describes the experiment apparatus. Section 4 and Section 5 present the details of the two experiments used to empirically test our approach. We conclude, in Section 6, with the discussion of limitations and future directions.

## 2. Background and Related Work

### 2.1. The Psychology of Emotional Facial Expressions

Facial expressions serve no biological survival purpose, but rather have been retained solely due to their importance in social interactions and the expression of emotion [26,27]. Facial expressions are the result of the movements of facial muscles groups present around the natural orifices. Furthermore, these expressions also involuntarily portray people’s inner emotional and cognitive states, even in isolated private settings [28] or with people who are socially challenged [29].

Facial expressions are the voluntary and involuntary movements that occur when facial muscles are engaged. They are a rich source of non-verbal communication and display a vast amount of emotional and cognitive information [30]. It has been well established that there are seven universal/basic facial expressions, namely, Joy, Sadness, Fear, Disgust, Anger, Contempt, and Surprise [31]. A person’s facial expressions, along with the situational context, allows an observer to infer the emotional state of the person. However, the accuracy of inferring emotional states from facial expressions of derived/complex emotions (e.g., frustration, boredom, confusion, awe, etc.) is lower compared to that of universal emotions. Current standards such as Facial Action Coding System (FACS) classifies facial expression based on the facial muscle movements from 44 action units (AU) [32]. FACS aids in the accurate and precise measurement of a wide range of facial expressions.

Emotions, similar to other psychological activities, are generally believed to stem from the underlying neural activities [33,34]. A study simultaneously monitored the neural activity with an electroencephalogram (EEG) and the facial behavioral responses with facial electromyography (fEMG) when the participants were exposed to emotional visual stimuli [35]. It appears that facial behavioral responses are caused by specific neural activities. However, the appearances of signals are usually not concurrent (with time delays), and some of the recorded neural activity did not correlate to facial responses.

Various studies, using either manual or automated visual analysis, have shown that emotional and cognitive loads appear to be conveyed via subtle facial expressions [10,32,36]. Additionally, a person’s expressions are voluntarily exhibited throughout various social, as well as isolated, contexts, that reflect upon their emotional and cognitive states.

### 2.2. Wearable Expression and Cognitive Activity Detection: Related Work

As already explained, facial expression analysis is a mature research field; however, only in as far as using stationary cameras to capture the facial expressions is concerned [9,10,11]. In this paper, we target a wearable system to which such video-based approaches are not directly applicable and in which much less successful work exists.

The main wearable approach to muscle activity monitoring is myography. It measures muscle contractions from their intensity and velocity. Currently, the most widely used myography method is electromyography (EMG), which measures the skin voltage potential change caused by the muscle movement. In [37,38,39,40], facial EMG are used to detect expressions with electrodes placed on participants’ forehead, temple or cheeks.

In [41], EMG electrodes are embedded in a mask to monitor racing drivers’ emotional states in simulated conditions. In [37], an unobtrusive device positioned at the temple area with EMG electrodes is demonstrated to be able to detect between neutral, smiling and frowning. Kwon, et al. [42] have used an array of electro–dermal activity (EDA) sensors combined with a small camera focused on the wearer’s left eye to detect emotional expressions. Inertial measurement units (IMU) placed inside the ear have also been studied to detect frowning and smiling in [43]. In [25], 17 photo reflective sensors are integrated into smart glasses to monitor the proximity between the facial skin and the eyewear frame.

EEG methods has also shown significant correlation with challenging tasks (*p* = 0.001) in [44]. Ren, et al. showed blink rate and blink rate variability have significant correlation with cognitive load (*p* < 0.05) under 1-back and 3-back tests in [45]. In [46], commercial smartwatches are used to measure office workers’ physical activities and predict their cognitive load.

### 2.3. Textile Mechanomyography

However, EMG requires electrical coupling between the sensing hardware and the body. The facial skin region is susceptible to long term abrasion and micro-current (e.g., electro-myo-stimulation). Moreover, EMG applications often require the use of electrolyte gels [47].

We investigate pressure force mechanomyography (MMG) as a possible solution to unobtrusively monitor facial muscle activities. Surface pressure force MMG measures the change of surface force with sensing elements mechanically coupled with the skin. MMG measured by a piezoelectric accelerometer has been compared with EMG to measure the activity of the first dorsal interosseous muscle of the hand in [48], with both EMG and MMG demonstrating similar capabilities of measuring the muscle contraction activities. In [49], probes constructed from force sensitive resistors and custom-designed mechanical structure has also been used to acquire the MMG signal. Textile pressure mapping (TPM) sensors measure the surface pressure change of the sensing fabric [50]. A study has shown flexible wearable garment embedded with TPM, which is coupled mechanically with the body, can be used to measure muscle surface activity. The Expression Glasses from [24] also uses a form of surface pressure MMG. In the Expression Glasses, two piezoelectric sensors are fixed by a rigid visor at the eyebrows to detect lowered or raised eyebrows.

### 2.4. Challenges

We propose an approach that aims to bridge the gap between the above-mentioned current state-of-the-art and a usable unobtrusive wearable device for facial expression recognition. In the process, we had to overcome some challenges as mentioned below.

#### 2.4.1. Anatomical Sensing Challenges

The high variability in shapes of foreheads depending on ethnicity and gender makes it hard to design a sensor that fits universally and can accurately sense forehead anatomical features. The variation in shapes is attributed to different degrees of slopes of the forehead and the presence of idiosyncratic bumps on the forehead. Therefore, sensors with sparse spatial granularity such as force sensors from the Expression Glasses in [24] or the abandoned early prototype in this work shown in Figure 1a cannot sufficiently capture the full range of muscle motions used in displaying emotional facial expressions.

#### 2.4.2. Psychological Challenges

There is huge variability in how different people express their emotions through their facial expressions. This is true not only for the seven universal expressions (joy, surprise, anger, fear, disgust, sadness, neutral) but even more so for complex negative expressions like frustration. Monitoring such negative emotions is important for providing user-specific support and encouragement during challenging tasks. Individual differences occur not only in the static versions of each facial expression but also in the temporal dynamics of how one arrives at a given facial expression.

## 3. Apparatus

We used the same smart fabrics as in [51] (Sefar AG). As shown in Figure 1c, the TPM sensor is essentially two layers of electrode stripes forming a two-dimensional array to measure the volumetric electrical resistance of a layer of carbon-polymer fabric (CarboTex^®^) in the middle. CarboTex’s local resistance decreases as more pressure is applied. A 10 × 2 TPM array with 1.5 cm pitch between sensing points was sewn into a headband (UnderArmour^®^). The textile sensor was covered with medical fabric tape. Since the TPM sensor is not stretchable, the part of the headband equipped with the sensor loses its elasticity. To compensate, we cut the headband at the opposite side of the sensor and extended its girth with a 10 cm elastic ribbon band. All of the participants reported no noticeable difference in the feeling between the instrumented and the original headband.

The smart fabric sensor brings several advantages. First, the soft sensing fabrics are covered with normal textile bands that are meant to be placed on the skin, removing skin irritation concerns. Second, the array can easily cover an area with multiple sensing points compared to using thin-film based FSRs. Prior to using the smart fabrics, an earlier prototype was also produced with four FSRs placed evenly along a line in the headband (Figure 1a). However, since the forehead region is an irregular-shaped contour, and there are bone structures underneath, the FSRs are prone to be positioned on high-pressure points (such as a corner of the bone structure) or low-pressure points (such as the more ‘flat’ area in the region) and thus fail to produce effective pressure MMG signals. This issue was not reported in the Expression Glasses in [24]. We suspect it is because of the rigid eyewear frame from that study helps to restrict the position of the piezoelectric sensors. However, with a soft headband, the said problem becomes more prominent. Whereas, a continuous TPM matrix is less prone to such garment sliding problems. Thus we have abandoned the 4×FSRs prototype.

A dsPIC32 microcontroller (Microchip^®^) with integrated 12-bit analog-digital converters (ADC) is used to scan the TPM array on a custom-designed printed circuit board (Figure 1d). The measurements are sent to a computer (Dell^®^ XPS 9550 i7) via a UART-to-USB bridge to ensure an optimal and consistent sampling rate. A Qt application written in C++ receives, parses, visualizes, and logs the pressure map at 66.7 frames per second or 15ms interval (the values are direct ADC output code). In this study, each of the two experiments required a media stimuli program. The stimuli programs and the data recording are run on the same computer so that the data received from the TPM and the media stimuli timestamps were synchronized. Further data analysis was carried out in Matlab^®^.

As in Figure 1b, the headband covers the skin directly above the eyebrows and under the hairline. For our purposes of detecting facial expressions, the relevant action units were AU1, AU2, and AU4. These three action units are connected to eyebrow movements and can be detected by our headband. The same apparatus was used in both experiments. For hygiene purposes, immediately before and after wearing the headband, the participants cleaned their forehead with provided antiseptic.

## 4. Experiment 1: Instructed Mimicking

The first experiment evaluates Expressure’s ability to distinguish expressions. We used an established psychological method [52] where participants were required to mimic particular expressions or emotions depicted in given images.

### 4.1. Participants

Twenty students (13 males, 7 females; age range 19-33) from the Technical University of Kaiserslautern, Germany, participated in this experiment. The participants were naïve to the experiment’s nature. All participants had a normal or corrected-to-normal vision. The participants reported not suffering from emotional disorders such as alexithymia and were able to express and identify their own and other’s emotions normally. The participants gave informed consent in accordance with the policies of the university’s Committee for the Protection of Human Subjects, which approved the experimental protocol. With their signed consent, their experiment was also video recorded, by a camera positioned behind the laptop screen on eye-level.

### 4.2. Experiment Design

Every participant was seated in an unoccupied single-person office with minimal distractions. They were shown visual stimuli of specific facial expressions with a media program and were asked to mimic the displayed expressions. The participants proceeded by clicking a single button with changing words on the user interface. The button shows ‘next’ by default, when clicked a stimulus was shown and the button changed to ‘record’. The participants mimicked the expression shown in the stimulus, and then click ‘record’. Then the program counted two seconds, during which, the participants tried to hold the expression and the button was disabled. After the two-second countdown, the stimulus disappeared, and the participants relaxed their facial muscles to their neutral face. Then after another second, the button changed back to ‘next’ and the participants proceeded to the next stimulus. The program logged the stimuli indexes and the timestamps when the participants clicked ‘next’ and ‘record’, as well as the end of the two-second countdown.

Based on the visual information’s complexity, the expression stimuli were categorized into ‘basic figures’ and ‘emotional portraits’ as shown in Figure 2. The basic figures consisted of either just a pair of lines for eyebrows or the eyebrows embedded in a simple cartoon face. The emotional portrait were actual photos depicting the associated facial expression. The experiment consisted of 5 repetitions of the stimulus sequence with 32 basic figures and 28 emotional portraits. It took 40 to 60 min to complete the entire experiment. Overall, every participant contributed 300 samples, and 6000 samples were recorded from 20 participants in total.

The basic figures were taken from [52]. Every figure was shown on the screen with a text label {‘neutral’, ‘lower eyebrow’, ‘raise eyebrow’} corresponding to the eyebrow status. There were two groups: the first group had three figures illustrating only the eyebrows, the second had five figures with a complete cartoon face with eyebrows and lip variations. The two groups were repeated four times each, and the order of appearance was randomized within each repetition by the program. For the data analysis, they are categorized into three labels as the eyebrow status {‘Neutral’, ‘Lower’, ‘Raise’}.

The emotional portraits were selected from the Warsaw set of emotional facial expression pictures (Warsaw Photoset) [53]. The Warsaw Photoset has 30 non-professional actors each showing seven emotions: {‘Neutral’, ‘Joy’, ‘Disgust’, ‘Fear’, ‘Anger’, ‘Sadness’, ‘Surprise’}. Every photo had been graded with other volunteers with measures including agreement, purity, and intensity, and also analyzed by a certified FACS coder to quantify the activation of facial action units. We first visually picked 8 actors from the Warsaw set whose 7 emotions are vastly distinct. For example, some actors’ ‘Anger’ and ‘Surprise’ were similar and hence confusing to mimic even with text labels. We further pared down the set to four actors (two males and two females) based on the measurements from the paper [53]. We felt that the facial expressions were the most distinguishable from these actors: JS, KA, KM, KP. Examples are shown in Figure 2b. The actors were shown to the participants in a consecutive order, and the seven expressions for a given actor were randomized with the corresponding emotion text label. We displayed 28 stimuli.

### 4.3. Data Analysis

First, the data is segmented as shown in Figure 3. To cover the transition between the participant’s neutral expression and the mimicking period, the window is wider than the actual ‘record’ period as shown by the program. As the transition to the mimicking expression happens shortly before the pressing of ‘record’, we chose a window starting from the middle between when they pressed ‘next’ (they saw the stimuli) and ‘record’ (they affirmed the mimicking). The end of the window is defined as 1 s after the end of the ‘recording’ period.

Figure 4 shows analysis process flowchart. The sensor data has three dimensions: a time sequence of 2D mappings (frames), written as F(t). A frame has 2×10 ‘pixels’: F(t)={p(x,y,t)|x∈[1,2],y∈[1,10]}. To extract features, we elaborated upon the spatial features from the studies where similar textile pressure is used to monitor quadriceps [51] and ball impact on a soccer shoe [54]. First, in a time window t∈T, key frames $KF_{n}$ are calculated by operating on the time domain:per pixel average of all frames:
(1)$$K\;F_{1}=\frac{1}{\left|T\right|}\sum_{t}^{\left\{T\right\}}F\left(t\right).$$sum of per pixel differences:
(2)$$K\;F_{2}=\sum_{t}^{\left\{T-1\right\}}\left(F\left(t+1\right)-F\left(t\right)\right).$$sum of positive or negative values of per pixel differences:
(3)$$K\;F_{3}=|\sum_{t}^{\left\{T-1\right\}}\left(\left(F\left(t+1\right)-F\left(t\right)\right)>0\right)|$$
(4)$$K\;F_{4}=|\sum_{t}^{\left\{T-1\right\}}\left(\left(F\left(t+1\right)-F\left(t\right)\right)<0\right)|.$$the frame which has the maximum mean pixel value as $K\;F_{5}$the frame with the minimum mean value as $K\;F_{6}$the frame with the maximum standard deviation from the stream as $K\;F_{7}$the per-pixel average of the frames, whose pixel value is greater than the frame pixel average:
(5)$$K\;F_{8}=\frac{1}{\left|T\right|}\sum_{t}^{\left\{T\right\}}\left(F_{p}\left(t\right)\right)$$
(6)$$F_{p}\left(t\right)=\left\{\begin{array}{cc} p(x,y,t) & \mathrm{if}\;p(x,y,t)\geq mean(F(t))\\ 0 & \mathrm{if}\;p(x,y,t)<mean(F(t)) \end{array}\right.$$

Key frames represent the pixel-wise changes during the time window. Examples are shown in Figure 5. Image moments (3 central moments plus Hu’s seven moments [55]) are applied on every key frame as features. We computed 80 features.

Three classifiers were compared: K-Nearest Neighbor (KNN), cubic kernel Support Vector Machine (SVM), and Ensemble bagged trees (TreeBagger). In all evaluations, five-fold cross-validation was performed on all participants’ combined data.

### 4.4. Cross-Validation of Detecting Facial Expressions

In the evaluation of this experiment, we define three different classification goals as three modes:Mode 1: three eyebrow states {Neutral, Lower, Raise}Mode 2: seven emotions {Neutral, Joy, Disgust, Fear, Anger, Sadness, Surprise}Mode 3: emotion groups based on eyebrow {{Neutral, Joy, Sadness},{Anger, Disgust},{Fear, Surprise}}

Since the feature distribution of three eyebrow motions and seven emotional expressions are different, we first decided the best performing classifier for each case. For Mode 1, the Neutral class has 800 data samples while the other two (Lower and Raise) each have1200 data samples. To ensure balanced cross-validation, we combined the Neutral class (400 data samples) from Mode 2 after confirming from the video recording that the two appear visually identical. Considering an average chance level of 0.333, the accuracy of three classifiers are KNN 0.762, SVM 0.828, TreeBagger (200 learners) 0.774. For 7 emotional expressions, with the chance level of 0.143, the accuracy of three classifiers are KNN 0.334, SVM 0.381, TreeBagger (400 learners) 0.373. The confusion matrices of both modes with the best performing classifier (SVM) are shown in Figure 6a.

Mode 1 confusion matrix shows that every eyebrow movement can be recognized with around 0.8 precision, well above the 0.333 chance. In the confusion matrix of Mode 2, there are more miss-classifications. For example, miss-classification occur often among Neutral, Joy and Sadness, but not between these three classes and the remaining classes. The precision of every class is still above the chance level of 0.143. Both confusion matrices from each mode have equal or similar F1 score and Accuracy, suggesting balanced precision and recall.

In Mode 2, it is obvious that the miss-classifications are concentrated at certain combinations of classes: {Neural, Joy, Sadness}, {Anger, Disgust}, {Fear, Surprise}. The class combinations out of these three groups are very well separated. When choosing our stimuli from the Warsaw Photoset, these groups were also difficult to distinguish at times. In the basic emotion wheel proposed by Plutchik [56], {Anger, Disgust} and {Fear, Surprise} are adjacent emotions and thus similar to each other. Therefore, the miss-classifications within these groups could be contributed by the similarity in the nature of these emotions. From the eyebrow movements, these groups largely represent, although not identical to, Neutral, Lower, and Raise eyebrows. Therefore, the emotions from said groups are bundled into three complex classes as Mode 3. Since the {Neutral, Joy, Sadness} group had 400 more data samples than the other two, 400 samples were randomly removed from this group to ensure every group had an equal amount of 800 samples. The confusion matrix is shown in Figure 6c.

### 4.5. Cross-Mode Validation

The confusion matrix of Mode 3 has lower, but similar recognition results compared to that of Mode 1. Essentially Mode 3 combines complex facial expressions that convey certain emotions via similar eyebrow states; but in Mode 1, the participants performed simpler expressions focused on the eyebrows. We examine how a classification model trained with basic actions can perform in such complex expressions. Therefore we used the trained SVM classifier from Mode 1 to predict the groups from Mode 3. Thus the Mode 3 groups are considered identical to the basic eyebrow motions: {Neutral:{Neutral, Joy, Sadness}, Lower:{Anger, Disgust}, **Raise**:{Fear, Surprise}}. As the result shows in Figure 6d, the accuracy of 0.743 is also well above the chance level and only a few percent less than Mode 3. This suggests that the Expressure method can be used to analyze the forehead region motions in complex facial expressions.

## 5. Experiment 2: Cognitive Loads

The second experiment aims at evoking reactions under cognitive loads with Double N-back tasks. N-back is a task commonly used in cognitive science to evaluate participants’ working memory [57,58], where participants need to identify repetitive patterns relying on their short-term memory. In cognitive load studies such as [45], N-back has proven to be an effective tool for inducing different levels of cognitive load.

### 5.1. Participants

Twenty students (aged between 21 and 28) from the Technical University of Kaiserslautern, Germany were recruited for this experiment, following the same recruiting and ethical procedures as Experiment 1 in Section 4.1. There were only two overlapping participants with the pool of participants of Experiments 1 and 2. The snapshots shown in Figure 9 were with explicit consent from the participants.

### 5.2. Experiment Design

We used the program Brain Workshop (BW, brainworkshop.sourceforge.net) to perform the N-back task. BW offers several modes of N-back, including multiple types of stimuli. In our experiment, we used the double stimuli mode of position and audio. An example of the program is shown in Figure 7. The participant saw a blue block appearing randomly in a 3 × 3 grid; at the same time, they also heard the audio of a random letter. The position and audio stimuli appear simultaneously for 0.5 s, every 3 s. One complete test round consisted of 21–24 pairs of stimuli (BW automatically decides the number of stimuli based on the current performance). Our further evaluation is based on the small time window between every two stimuli. In total, 20 participants contributed 4266 such window samples.

A stimulus is considered positive if it is the same as the N-th previous stimuli; otherwise, the stimuli are negative. The participants input their answer by pressing the key ‘A’ and ‘L’; ‘A’ indicates the block position matches the N-th previous position, and ‘L’ indicates that the heard letter matches the N-th previous letter. Otherwise, the participant is considered to perceive no match (negative input). If the positive input equals the stimulus, the program indicates by showing the corresponding text (position or audio) in green; if the positive input is wrong, the corresponding text turns to red; otherwise, the texts remain black. The program logs the time and label of each stimulus and input.

Before the actual recording with the BW program and in order to familiarize the participants with the task, every participant runs a single-stimulus 2-back test with written letter stimuli for three practice sessions. After the practice sessions, they wear the Expressure headband and complete nine sessions of double N-back tasks. From our initial trials, after several sessions the participants tend to get used to the duo 2-back tests, making the tests less challenging. Therefore, we increased the difficulty of the 4th sessions to duo 3-back. Because duo 4-back is too difficult for any beginner, it is not practical to further increase the difficulty in our experiment. Instead, we lower the difficulty back to duo 2-back from the 7th session. The last three sessions will still appear challenging to the participants because of the change in tasks from 2-back to 3-back and again to 2-back. Thus the test sequence is: three sessions duo 2-back, three sessions duo 3-back, and three sessions duo 2-back. In the psychology doctrine, a score is calculated as the measure of the participant’s performance in an N-back task as:(7)$$\frac{\mathrm{sum}\;\mathrm{of}\;\mathrm{all}\;\mathrm{correct}\;\mathrm{positive}\;\mathrm{inputs}}{\mathrm{sum}\;\mathrm{of}\;\mathrm{positive}\;\mathrm{stimulus}\;+\;\mathrm{sum}\;\mathrm{of}\;\mathrm{false}-\mathrm{positive}\;\mathrm{inputs}}$$

The N-back score of every participant and every session is shown in Figure 8, where it is clear that as the participants repeat the same test several times, their score increases. We can also observe that changing the difficulties in phases has successfully readjusted the participants’ familiarity with the task and therefore halted the increasing score pattern. This is a desired outcome because the purpose of the experiment is to evoke cognitive-related reactions, which are more likely to occur for difficult tasks. The median N-back score of the entire experiment from all participants was 0.2857. Between every session, the participants could take a break for as long as they needed. They were also required to lift the headband and put it back, with the goal of introducing more sensor position variations.

To evoke natural cognitive related responses, it is important the participants did not know the purpose of the headband. Otherwise, they may move or refrain from moving the forehead muscles [39]. Therefore, we avoided giving away any information on the actual sensing technology. Participants were only told that the headband operates on very low current and is carefully isolated so there are no safety concerns.

We manually annotated the participants’ facial reactions from the video recording of the experiment with the text description as the video transcript. Some participants, such as Participant 1 as shown in Figure 9a, have very vivid and active facial expressions under the cognitive load induced by the test. However, there are also participants who do not show visible expression changes even during sessions with low scores, such as participant 3 in Figure 9b. The photo is from a session with 0.18 score, and the participant kept the same expression during the whole session.

### 5.3. Data Analysis Method

The data analysis process is shown in Figure 10. Instead of measuring the participants’ working memory, we used N-back task as a medium to modulate cognitive load and evoke emotional reactions such as frustration, confusion, and confirmation. We analyze the N-Back task on a fine-grain per stimulus basis. A time window is defined between every two consecutive stimuli. Within a window, we compute the same features as in Experiment 1. Since the users’ reactions can be voluntary, complex, and subtle, there are not enough samples to form specific groups for classification purposes. We instead use the stimuli and input of the N-Back task to form different conditions. These conditions indicate whether the participants are experiencing cognitive challenges. Therefore, the evaluation goal is to see if there is a difference in the features’ distribution under different conditions.

Figure 11 shows the signal from the Expressure headband as average pressure, with the real-time stimuli and inputs, in a session of double 2-back. The data is further segmented into windows by adjacent stimuli. From the figure, it can be seen that sharp peaks or changes from the average pressure usually appear with positive stimuli or inputs, or wrong inputs, such as window {2, 5, 7, 8, 9, 19, 22}. There are also times that peaks occur with no positive stimuli or inputs, such as windows 1 and 3. Moreover, positive stimuli or inputs may not cause any change in the average pressure data, such as {13, 14, 15}. From our observation and the video recording transcript, the participants do not always have an observable response to positive stimuli or faulty answers. However, repeated wrong answers usually lead to expressions of frustration. Therefore, we aim to find statistical correlations between every window of stimuli/input and the participants’ facial expressions captured by our sensor.

We defined eight combinations of the stimulus and input conditions in Table 1. These stimulus/input conditions were external objective indicators that could reflect the participants’ internal subjective cognitive activities. To further illustrate these conditions, they are shown in the form of truth tables in Figure 12, where the golden cells indicate the conditions, and the purple cells the opposite-logic of these conditions. Take the first truth table in Figure 12 for example, the golden cells indicate any positive stimuli in the window (either the audio or position stimulus is 1); while the purple cells indicate the opposite logic, that both stimuli are negative.

The average pressure is not sufficient to represent the complex forehead movements. Therefore, we use the same features from Experiment 1 (image moments on key frames) on every between-stimulus window. In this experiment, the cognitive-related reactions from the participants are voluntary and complex. Prior to the experiment, they are not aware that the headband is measuring the motion of their forehead muscles. For many participants, these cognitive-related reactions include shaking head, lip motion, biting lips, etc. without obvious changes in the eyebrows. In many instances, change in the pressure sensor signal is visible, but the facial expression cannot be categorized through video into any of the classes from Experiment 1. The two experiments also only shared one common participant. Therefore, we cannot use the classification models from the first experiment, where the expressions were mimicked under clear instructions and visual stimuli, to perform prediction in the scope of Experiment 2. Our evaluation aims to see if there is any statistical significance in the features’ distribution from each stimulus/input condition.

Each feature was normalized so that the mean equals 0 and the standard deviation equals 1. We sorted the windows according to the conditions and the corresponding opposite-logic conditions. Thus, under each condition, there were two groups of windows, each window has 80 features. For every feature, we constructed a pair of histograms, one for the windows under the condition (colored as golden), and another for those under the opposite condition (colored as purple). They were in the range of [–2, 2] with 26 bins. If there is no significant difference between the pair of conditions, the two sides of the histogram pairs shall appear to be similar (symmetric); otherwise, the two sides shall present clear divergence (asymmetric).

The pair of histograms can be compared with various histogram similarity measurements. We follow the method described in [59] of comparing the consistency of two histograms. To have a balanced comparison, each bin’s value is divided by the sum of the corresponding histogram values, then multiplied by half the amount of total windows from Experiment 2, so that the two histograms have an equal sum, and together add up to the total amount of windows. We used the Chi-Square test on the two distributions, to calculate the Chi-square ($\chi^{2}$) critical value (CV). A bigger Chi-square CV value indicates the distribution are more distinct from each other. For every condition, we choose the feature that gives the highest CV value as the most significant feature for the condition. With the degree of freedom of 25, we can also calculate the *p*-value of *p* ($\chi^{2}>$ CV) and determine the statistical significance.

### 5.4. Correlating with Short Time Window Cognitive Activities

From the histograms in Figure 13, asymmetry can be observed in every pair. The null hypothesis here is that the two histograms from a pair are sampled by equal probability (randomness) from the same distribution. Put differently, the null hypothesis assumes there is no correlation between the data distribution and the two logically complementary conditions.

Table 1 lists the Chi-square critical value (CV) and *p*-value of every condition. For all conditions, feature 48 has the highest CV value. We consider the common statistical significance standard for the *p*-value of smaller than 0.05. Condition 4 (any positive correct input), 6 (any positive input or stimulus, both inputs match stimuli), 7 (any input matches stimulus), 8 (user sees a red feedback from the test program) has a *p*-value of 0.0100, 0.0002, 0.0000, 0.0100 respectively. This means that we cannot accept the null hypothesis. Hence, there is a significant correlation between the shown features’ distribution and the corresponding stimulus/input conditions. Condition 3 (both inputs equal stimuli) also has a *p*-value of 0.0900 which is close to the significance threshold.

The similarity of these conditions is that the separation is based on matching stimuli and inputs. The correct and positive inputs are also used to calculate the typical N-back score generated by N-back tasks, as a measure of the participant’s working memory and cognitive load.

### 5.5. Predicting Cognitive Loads during Longer Periods

We also compared the sessions with higher and lower than median N-back scores. For every feature, we sorted the windows into two histograms according to the session’s N-back score. Thus each session contributes 21–24 points to be counted in the histogram. Each feature results in a pair of histograms, separated by the total median N-back score. Since the window amounts are similar, these histogram pairs are not normalized or balanced. Thus, the null hypothesis, in this case, is that the two histograms from a pair are sampled by equally random probability from the same distribution. The null hypothesis can also be stated as there is no correlation between the data distribution and whether the session’s N-back score is above or below the total average.

Then we compute the CV value for each pair of histograms. The pairs with the top two highest and two lowest CV values are shown in Figure 14. We can observe that the feature distribution is visually distinct for the sessions with higher and lower than average N-back scores. Top CV values with 25 degrees of freedom result in 0.0000 *p*-values. The feature with the smallest CV value (the least significant correlation) result in 0.03 *p*-value, also below the 0.05 threshold. Thus, we can conclude that there is significant correlation between the distribution of all the features and whether the session has an N-back score above or below the median.

To transfer the statistical correlation into classification results, we performed binary-class cross-validation, with the two classes being higher or lower than the median score. From all windows of a session, the average, standard deviation, range, and kurtosis were calculated for each feature. The outcome was taken as the new feature of the session. We then used the Classification Learner from Matlab to find the best-performing classifier, which was Subspace KNN. Since some participants were not particularly facially expressive during the test, we divided the participant pool into the 10 most expressive participants and the 10 least expressive participants (Figure 15). The confusion matrices are shown in Figure 15. We can also observe that the accuracy values are all above 0.66, and the accuracy among the more expressive participants is 0.767.

Our findings from Experiment 2 suggest that:

(1) Changes in facial expression and head motion are present when the participants are under cognitive load.

(2) During a long period of time, segmented by periodic windows of a few seconds, the distribution of the short windows’ features had a statistically significant correlation with the wearer’s cognitive activity with *p*-value well below 0.05.

(3) The said correlation also leads to the prediction of whether the session score is above the median with up to 0.767 accuracy from the more expressive participants.

## 6. Conclusions, Discussion and Future Outlook

In this paper, the presented work has provided initial evidence that the proposed Expressure approach can be a solution, to unobtrusively monitor facial expressions and cognitive loads with a soft and comfortable wearable device. The device is in the form of a sweatband, using smart textiles to perform surface pressure mechanomyography, in order to detect the forehead muscle movements. This study argues for the usage of wearable devices for emotion recognition to overcome drawbacks from that of computer-vision and bio-electrical signal-based approaches. Compared to the bio-electrical signal based systems, our novel approach has the advantages of comfort, easy to be put on and taken off, and no skin-electrode contact requirement.

In this study, we have thoroughly described the architecture of this system and presented an initial evaluation with systematic psychological experiments. The first mimicking expression experiment validates Expressure’s ability to recognize facial expressions voluntarily copied from portraits. Based on the data of 6000 samples from 20 participants, we demonstrate the system’s ability to detect well-defined facial expressions. We achieved accuracies of 82% to classify among three eyebrow movements (33% chance level) and 38% to classify among seven full-face expressions (14% chance level). In the related work, computer vision methods can achieve precise expression classification, for example, in [15], the Facenet2expnet can classify seven emotions with 89% accuracy. However, apart from the camera setup, such methods benefit from multiple facial markers (lips, cheek, eyes, etc.); while our proposed method only uses the forehead region and is unobtrusive. EMG placed at the temples has shown to detect a smile with around 90% accuracy in [60]. EEG has also been studied for emotion detection with above 90% accuracy among four emotions in [61]. We argue that while EEG or EMG relies on stable electrode contact with the skin as mentioned in Section 1, the proposed Expressure approach does not require any electrical skin contact since the muscle activity is mechanically coupled with the pressure sensors.

The second experiment with N-back tasks has shown a statistically significant correlation between the features acquired by Expressure and the participants’ cognitive activities. Based on the data of 4266 window samples collected from 20 participants, we have shown that features calculated from the Expressure data on a fine time granularity of every three seconds, has significant correlations with the cognitive activity indicated by the objective N-back stimuli and inputs. The results have also shown significant correlations between the Expressure data and the N-back scores with *p*-values smaller than 0.05. Studies in the literature with blink rate and blink rate variability with 1-back and 3-back tests have also shown similar evaluation standards with smaller than 0.05 *p*-value [45]. In [44], EEG based method was revealed to correlate with challenging tasks with 0.001 *p*-value. We further used the Expressure features to train a classifier to predict whether the N-back score is above or below the average from our recordings. From the 10 most facially expressive participants, the classification accuracy was up to 76% (50% chance level).

Overall, our study has shown promising results of using our smart wearable sweatband to detect not only people’s facial expressions but also their cognitive loads by monitoring the dynamic facial expressions while performing a challenging task over a period of time.

With respect to our system design, we had to overcome anatomical sensing challenges due to the high variability of forehead shapes among individuals. Another challenge encountered has been to make the sweatband comfortable to wear for a longer duration. Moreover, individuals differ in the temporal dynamics of showing emotion through their facial expressions as well as in the static versions of each facial expression. This makes the coding and decoding of emotions from the pressure data recorded by our sensors a challenging task. Our long-term goal is to be able to monitor complex negative emotions such as frustration as it is important for providing customizable support and encouragement during cognitively challenging workloads.

It is well established that frustration negatively impacts job performance [3,62,63], therefore it will be ideal to develop a non-intrusive way to continuously monitor frustration in order to provide appropriate support and feedback. With Expressure, it would be interesting to perform further long-term user studies in scenarios such as learning, problem-solving, sports practicing, etc. The availability of a tool that can accurately measure such intricate mental details of a person could prove to be advantageous in many instances, such as by sports coaches to monitor the emotional states and cognitive loads of their athletes, driving, assessment of students’ cognitive loads by teaching staff, and situations that involve human-machine interaction. It could also prove to be important for monitoring negative emotions in children because they are more likely to give confirmatory responses socially acceptable to adults in order to appease them. A lot of information stands to be gained by accurately measuring the movements of facial muscles and developing a comprehensive model to classify them into various emotional states; information that could benefit various fields from artificial intelligence to sports management and coaching to psychological studies involving emotional states.

As part of the future work, we will extend our approach to include additional sensors (e.g., head motion, optical heart rate sensors) and to cover fore facial areas (e.g., through a glasses frame that would provide signals from the nose and side areas below the ears) in order to better translate human feelings for human–machine interaction and affective computing.

## Figures and Tables

**Figure 1 sensors-20-00730-f001:**
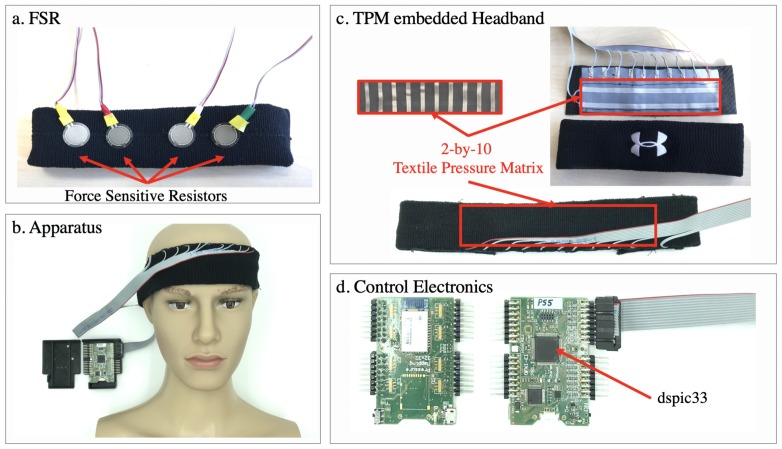
(**a**) Prototype with four force-sensitive resistors (FSR) that was rejected after comparing with the new approach. (**b**) The apparatus used for our experiments shown with the textile pressure mapping sensors embedded inside the headband. (**c**) The interior of the headband with a 2-by-10 textile pressure mapping (TPM) matrix, isolated from the user’s skin with textile tapes. (**d**) The electronics hardware, built around a dspic33 microcontroller.

**Figure 2 sensors-20-00730-f002:**
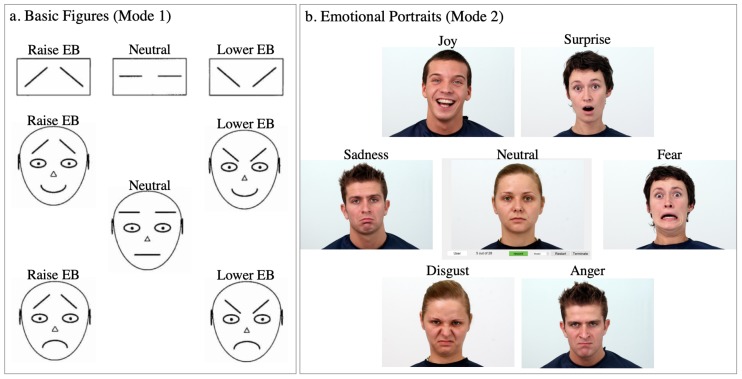
The visual stimuli during Experiment 1. (**a**) Basic figures are selected from [52]. (**b**) Emotional Portraits are selected from [53].

**Figure 3 sensors-20-00730-f003:**
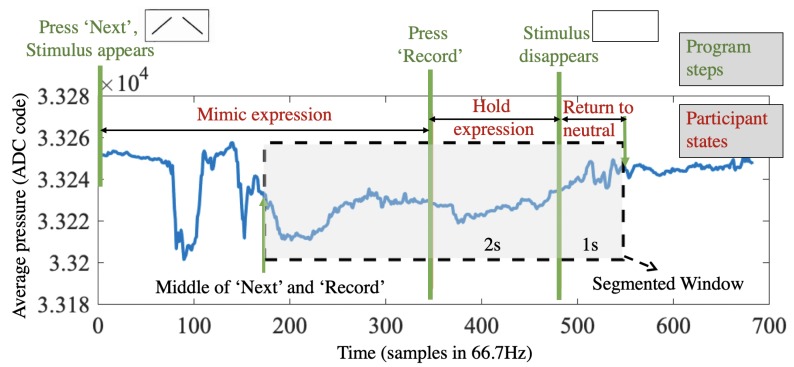
Window segmentation of Experiment 1, so that it can best cover the transition between the neutral expression and the mimicking period.

**Figure 4 sensors-20-00730-f004:**
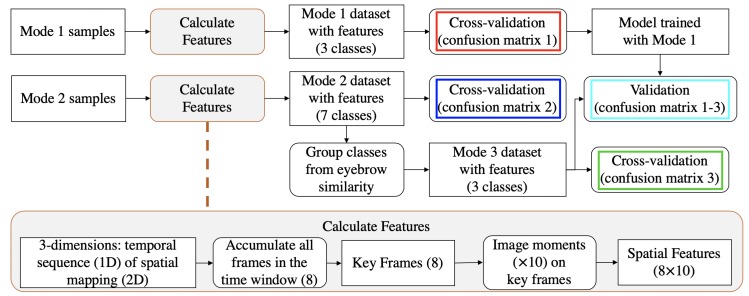
Flowchart of the data analysis process in Experiment 1.

**Figure 5 sensors-20-00730-f005:**
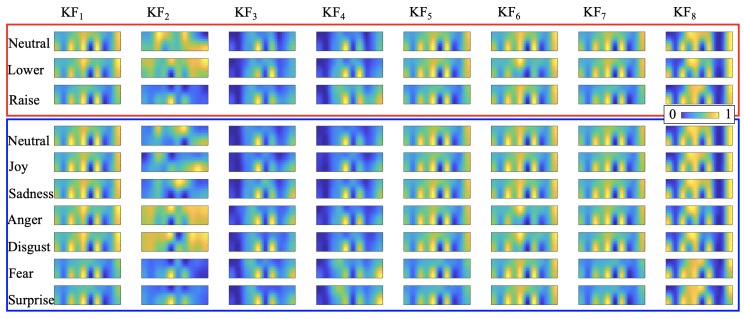
Key frames examples from a single participant (participant 6).

**Figure 6 sensors-20-00730-f006:**
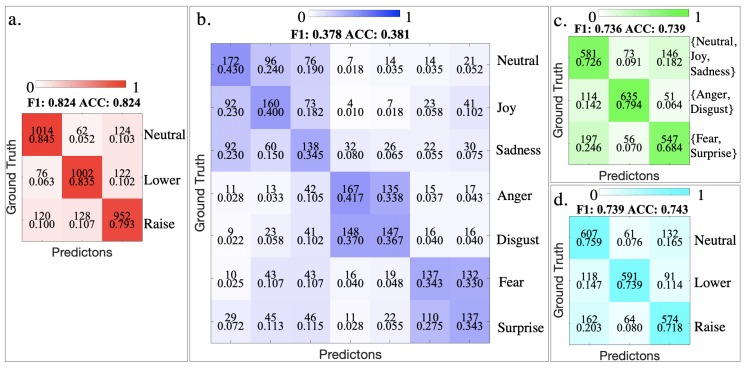
Confusion Matrix of Mode 1 (**a**), Mode 2 (**b**) and Mode 3 (**c**). (**d**) is the confusion matrix of a Support Vector Machine (SVM) classifier trained with data from Mode 1, tested on the data from Mode 3. The upper number in each cell is the sample count, the lower number is precision and false negative rate.

**Figure 7 sensors-20-00730-f007:**
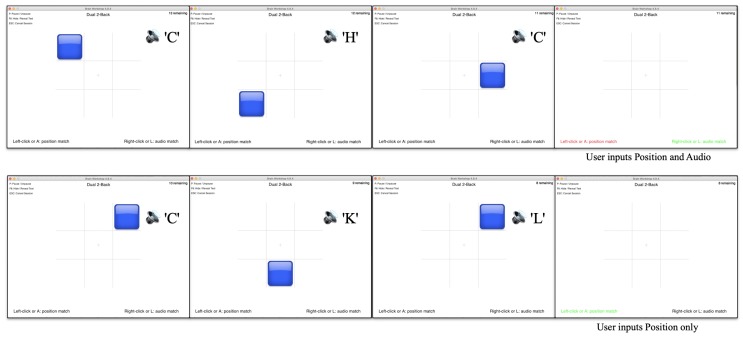
Example of the brain workshop running a dual 2-back test. The stimuli are shown for half a second, every three seconds. The participants need to indicate within the 3-s window if the position or audio matches the 2nd (or N-th) previous stimuli. The writings at the bottom of the program indicates whether participants’ inputs are correct by turning green or wrong by turning red.

**Figure 8 sensors-20-00730-f008:**
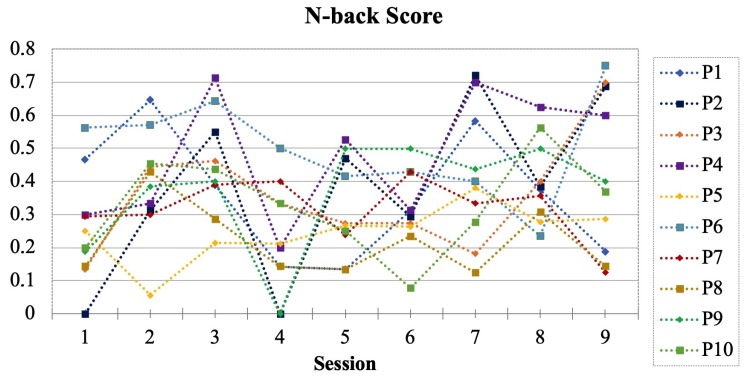
The N-back score of the 10 participants. A clear drop during sessions 4–6 (double 3-back) is visible.

**Figure 9 sensors-20-00730-f009:**
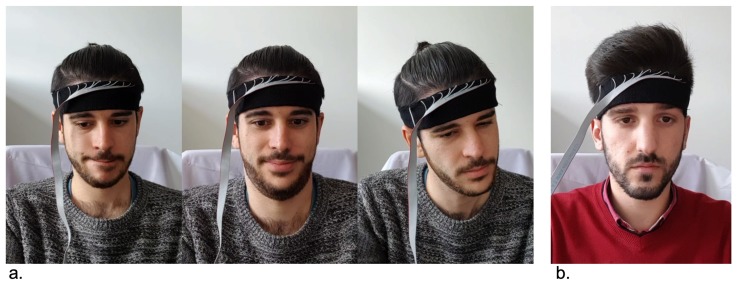
(**a**) Participant 1 has very vivid and active expressions under cognitive challenges. (**b**) Participant 3 has almost no changes of expression during the entire experiment, even during sessions with lower than average scores.

**Figure 10 sensors-20-00730-f010:**
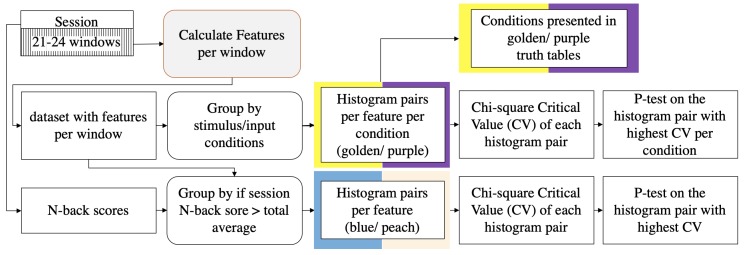
Flowchart of the data analysis process for Experiment 2. The ‘Calculate Features’ process is the same as the one in Experiment 1.

**Figure 11 sensors-20-00730-f011:**
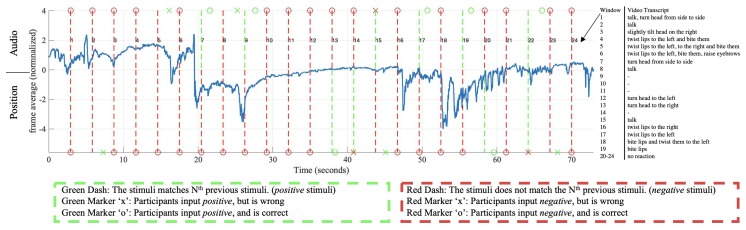
Signal examples of the double 2-back test from two participants. The signal plotted in blue is per frame average. The dash lines indicate the appearances of stimulus, with audio stimulus above and position stimulus below the zero line. The markers are user inputs as explained in the figure. A window is defined between every two stimulus (dash lines). The manual video transcript of every window is shown next to the plot.

**Figure 12 sensors-20-00730-f012:**
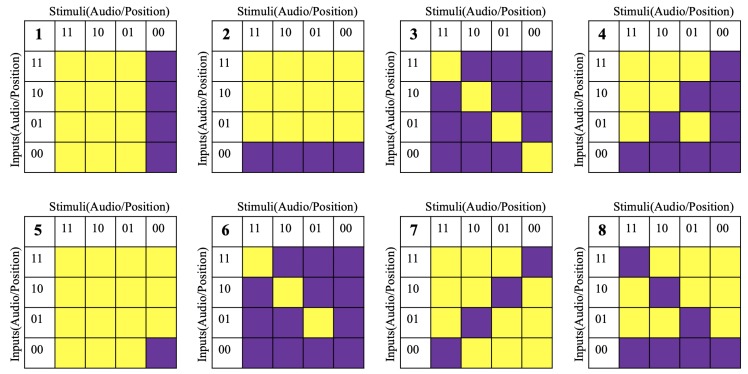
Truth table of the stimulus/input conditions. Golden cells are the conditions listed in Table 1, the Purple cells are the opposite logic.

**Figure 13 sensors-20-00730-f013:**
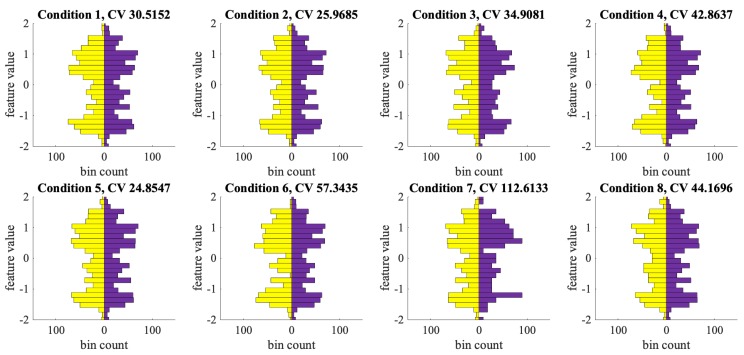
Histogram pairs of the most significant features under eight conditions. With less CV value (weaker statistical significance), the distribution pair would be more symmetric.

**Figure 14 sensors-20-00730-f014:**
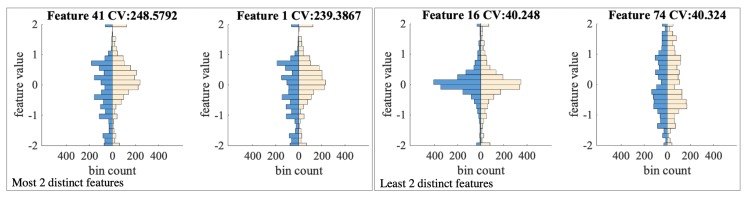
The histogram comparison of those windows with a N-back score above median (in blue) and not (in peach color). Two features resulting in the highest and two with the lowest CV values are chosen.

**Figure 15 sensors-20-00730-f015:**
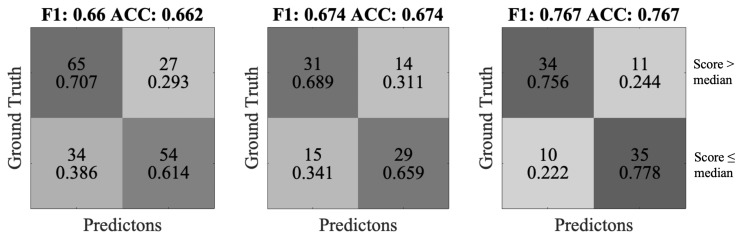
The confusion matrices of identifying whether the session score is above or below average. From left to right: all, 10 least expressive and 10 most expressive participants.

**Table 1 sensors-20-00730-t001:** The $\chi^{2}$ (critical value (CV)) and *p*-value ($\chi^{2}$ > CV) of the most significant feature (smallest *p*) for each stimulus/input condition, and the number of features (Num.) exceed the significance level.

No.	Stimulus/Input Conditions	$\chi^{2}$	*p*	Num.
1	Any positive stimulus	30.5152	0.2100	0
2	Any positive input	25.9685	0.4100	0
3	Both inputs equal stimuli	34.9081	0.0900	0
4	Any positive correct input	42.8637	**0.0100**	1
5	Any positive input or stimulus	24.8547	0.4700	0
6	Any positive input or stimulus	57.3435	**0.0002**	18
	and both inputs match stimuli			
7	Any input matches stimulus	112.6133	**0.0000**	68
8	Any positive input does not	44.1696	**0.0100**	4
	match stimulus (red feedback)

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
