# Peer review of "Expressure: Detect Expressions Related to Emotional and Cognitive Activities Using Forehead Textile Pressure Mechanomyography"

_sensors, 2020, doi:10.3390/s20030730_

Round 1

Reviewer 1 Report

Comments:

1) The authors in the background describe different techniques to detect emotions but they do not indicate the numerical results obtained by the other experiments

2) 20 participants per experiment are not enough for statistical analysis. It would be better to add more participants to the study

3) The authors should indicate the inclusion and exclusion criteria of the participants in the study

4) Conclusions section is very short and that section should contain the comparison of results between the proposed system and the literature

5) Some references are very old, and other are conference publications. The authors should add high quality references

Author Response

1) The authors in the background describe different techniques to detect emotions but they do not indicate the numerical results obtained by the other experiments

Thank you for the comment. We've added numerical discussions in the conclusion section with relevant literature. However we would like to emphasis that, to the best of our knowledge, our approach of detecting expressions with mechanomyography was not studied before.

2) 20 participants per experiment are not enough for statistical analysis. It would be better to add more participants to the study

Thank you for the comment. Indeed 20 is not a sufficient sample for statistical analysis. We want to clarify that the results presented for both experiments are not based on individuals, but rather small time samples from all participants combined: in Experiment 1, the participants have performed multiple repetitions with randomized sequences, in total 6000 samples, and in Experiment 2, the statistical analysis is performed on time window basis between every two stimuli, in total 4266 samples. We further explain the sample numbers accordingly in the experiment design section and highlighted in the resubmission.  Additionally, in our cited work with novel sensing modality for activity recognition, we’ve seen mostly between 5 to 30 participants. Studies with more participants are mainly dependent on computer vision or off-the-shelf devices, which are very easy to be scaled up. As we would very much like to, but with the journal’s 10-day major revision deadline it is difficult to arrange any further recordings.

3) The authors should indicate the inclusion and exclusion criteria of the participants in the study

Thank you for this comment, in response, we have added a new section of Participants in each of the experiments as section 3.1 and 4.1.

4) Conclusions section is very short and that section should contain the comparison of results between the proposed system and the literature

Thank you for the comment. We have re-written the conclusion section, which is highlighted in the resubmitted document.

5) Some references are very old, and other are conference publications. The authors should add high quality references

We appreciate this comment together with the other reviewer. We added 9 more recent papers and highlighted accordingly in the reference. Since emotion and expression are a well established psychology branch, we still feel all our existing references are worth mentioning.

Reviewer 2 Report

Authors present the Expressure system, which performs surface pressure mechanomyograpghy on the forehead employing an array of textile pressure sensors. This system could be used to detect emotions and cognitive states related facial expressions. This system uses a  soft and comfortable headband through smart textiles for monitoring the forehead muscle movements. Only, this manuscript must be improved considering the following comments:

1.-Authors must include more recent references (e.g., references between 2017 and 2020).

2.-Authors must add more discussions about results reported in Figures 11-15.

Author Response

1.-Authors must include more recent references (e.g., references between 2017 and 2020).

We appreciate this comment together with the other reviewer. We added 9 more recent papers and highlighted accordingly in the resubmitted pdf. Since expression and cognitive load are well established 

2.-Authors must add more discussions about results reported in Figures 11-15.

Thank you very much for this comment. We have added more explanations and discussions for experiment 2 in Section 4. We've also added further discussions in the conclusion section. We've highlighted these changes in the resubmitted pdf.

Round 2

Reviewer 1 Report

The modifications made by the authors are enough to publish the paper in the present form.

Reviewer 2 Report

Authors have improved their manuscript considering all the reviewer's comments. This manuscript is suitable to be published in Sensors.